# AN69 Filter Membranes with High Ultrafiltration Rates during Continuous Venovenous Hemofiltration Reduce Mortality in Patients with Sepsis-Induced Multiorgan Dysfunction Syndrome

**DOI:** 10.3390/membranes11110837

**Published:** 2021-10-29

**Authors:** Kuo-Hua Lee, Shuo-Ming Ou, Ming-Tsun Tsai, Wei-Cheng Tseng, Chih-Yu Yang, Yao-Ping Lin, Der-Cherng Tarng

**Affiliations:** 1Division of Nephrology, Department of Medicine, Taipei Veterans General Hospital, Taipei 11217, Taiwan; dadabim3520@gmail.com (K.-H.L.); okokyytt@gmail.com (S.-M.O.); mingtsun74@gmail.com (M.-T.T.); wctseng@gmail.com (W.-C.T.); cyyang3@vghtpe.gov.tw (C.-Y.Y.); linyp@vghtpe.gov.tw (Y.-P.L.); 2Faculty of Medicine, School of Medicine, National Yang Ming Chiao Tung University, Taipei 11217, Taiwan; 3Institute of Clinical Medicine, National Yang Ming Chiao Tung University, Taipei 11217, Taiwan; 4Center for Intelligent Drug Systems and Smart Bio-Devices (IDS2B), Hsinchu 30010, Taiwan; 5Department and Institute of Physiology, National Yang Ming Chiao Tung University, Taipei 11217, Taiwan

**Keywords:** critical medicine, multiple organ failure, intensive care unit, blood purification

## Abstract

Polyacrylonitrile (AN69) filter membranes adsorb cytokines during continuous venovenous hemofiltration (CVVH). Although high-volume hemofiltration has shown limited benefits, the dose-effect relationship in CVVH with AN69 membranes on severe sepsis remains undetermined. This multi-centered study enrolled 266 patients with sepsis-induced multiorgan dysfunction syndrome (MODS) who underwent CVVH with AN69 membranes between 2014 and 2015. We investigated the effects of ultrafiltration rates (UFR) on mortality. We categorized patients that were treated with UFR of 20–25 mL/kg/h as the standard UFR group (*n* = 124) and those that were treated with a UFR >25 mL/kg/h as the high UFR group (*n* = 142). Among the patient characteristics, the baseline estimated glomerular filtration rates (eGFR) <60 mL/min/1.73 m^2^, hemoglobin levels <10 g/dL, and a sequential organ failure assessment (SOFA) score ≥15 at CVVH initiation were independently associated with in-hospital mortality. In the subgroup analysis, for patients with SOFA scores that were ≥15, the 90-day survival rate was higher in the high UFR group than in the standard UFR group (HR 0.54, CI: 0.36–0.79, *p* = 0.005). We concluded that in patients with sepsis-induced MODS, SOFA scores ≥15 predicted a poor rate of survival. High UFR setting >25 mL/kg/h in CVVH with AN69 membranes may reduce the mortality risk in these high-risk patients.

## 1. Introduction

Acute kidney injury (AKI) frequently occurs among critically ill patients due to a high disease burden and polypharmacy. The rapid loss of renal function may lead to severe complications such as fluid overload, electrolyte imbalance, acid–base dysregulation, and uremic toxin accumulation. As this is compounded with concomitant systemic inflammation, these developments may affect other vital organs, resulting in multiple organ dysfunction [1]. Therefore, AKI and multiorgan dysfunction syndrome (MODS) exert negative effects on survival among patients that are in intensive care units (ICU) [2]. A renal replacement therapy (RRT) option for managing AKI and MODS, continuous RRT (CRRT), allows for steady fluid and waste removal and has become the preferred treatment option for critically ill patients with hemodynamic instability [3,4].

Regarding the transport of solutes in CRRT, convection is the process whereby solutes pass through the membrane pores of the filters and are dragged by fluid movement (ultrafiltration) under a hydrostatic transmembrane pressure gradient [5]. Continuous venovenous hemofiltration (CVVH) is a pure convective CRRT technique that may afford additional advantages in treating ICU patients by removing middle molecule-sized proteins such as cytokines [6,7]. CVVH uses large amounts of replacement fluid without dialysate and, therefore, the convective efficacy of solute clearance completely depends on the ultrafiltration rate (UFR). However, there is only limited evidence that high-volume hemofiltration improves outcomes in critically ill patients [8,9]. On the other hand, polyacrylonitrile (AN69) filter membranes have been shown to have a high capacity to adsorb cytokines (tumor necrosis factor (TNF)-α, interleukin (IL)-1β, IL-6, IL-8, and IL-10) and high mobility group box-1, a representative of damage-associated molecular patterns in experimental studies [10,11,12]. The membrane is negatively charged from methallylsulfonate, adsorbing cytokines via ionic bonding between the sulfonate groups and amino groups on the cytokines. Accordingly, CVVH with AN69 filter membranes and a high UFR appears to be the preferred strategy for removing circulating inflammatory mediators, particularly for patients with severe sepsis.

Ronco et al. indicated that an effluent rate of >35 mL/kg/h of CVVH was associated with better survival among patients with AKI necessitating dialysis (AKI-D) than was a rate of 20 mL/kg/h [13]. However, two pivotal randomized controlled trials (RCTs), namely the Acute Renal Failure Trial Network study [14] and the Randomized Evaluation of Normal vs. Augmented Level of Replacement Therapy study [15], have suggested that a higher intensity of CRRT does not reduce the mortality risk or improve renal recovery among patients with AKI. According to the Kidney Disease: Improving Global Outcomes (KDIGO) guidelines, the optimal CRRT dose is an effluent flow rate of 20–25 mL/kg/h [16]. However, in the two aforementioned RCTs, the participants who used CRRT were all treated with continuous venovenous hemodiafiltration, and in 50% of the patients who developed AKI-D, the cause was sepsis. Given that sepsis is the most common etiology of AKI and MODS in the ICU and is a strong predictor of in-hospital mortality, it is crucial to assess the effect of CVVH with a higher UFR and AN69 filters that have a high absorptive capacity on the mortality of patients with sepsis-induced MODS. We hypothesized that different target doses of CVVH may be required at various stages of critical illness, but this question has not been examined in prospective RCTs. Therefore, we conducted this multi-centered cohort study to assess the effect of standard UFR (20–25 mL/kg/h) versus the higher than guideline-recommended UFR (>25 mL/kg/h) on the patients’ survival of sepsis-induced multiorgan dysfunction.

## 2. Materials and Methods

### 2.1. Study Design and Data Source

We conducted this multi-center, register-based study between 2014 and 2015. The participants comprised of critically ill patients with sepsis that were treated with CVVH in the ICUs of 10 medical centers and 12 regional hospitals in Taiwan. Each participating site recruited individuals independently and uploaded their data to an online database http://www.caks.org.tw (last accessed 30 September 2019) for registration. To assure the data quality and accuracy, the senior physicians of each hospital audited the chart review process and were responsible for error correction. The study was conducted according to the guidelines of the Declaration of Helsinki, and the study protocol was approved by the Institutional Review Board (IRB) of the National Research Program for Biopharmaceuticals (Approval No. NRPB2014050014).

### 2.2. Participants

The patients were eligible if they were adults that were aged ≥20 years-old, diagnosed as having sepsis under the Sepsis-3 criteria [17], complicated with ≥2 organ dysfunctions (including AKI), and were subsequently treated with CVVH in ICUs. AKI was diagnosed according to the criteria that are presented in the KDIGO 2012 Clinical Practice Guideline for the Evaluation and Management of Chronic Kidney Disease, which is based on changes in the serum creatinine (sCr) from a baseline or a reduced urine output [16]. The baseline sCr level was defined as the average of all of the sCr levels of the previous discharge within 1 year of the current hospital stay or the outpatient clinic blood tests within 6 months. If both types of data were available, the outpatient data were used. If baseline sCr data were not available, the lowest sCr levels that were recorded during the current hospital stay were used. If these two approaches both failed (i.e., if there were no past admissions or clinic visits and if the nadir levels were equal to the first sCr values without subsequent changes), we used the Modification of Diet in Renal Diseases (MDRD) equation by assuming a baseline estimated glomerular filtration rate (eGFR) of 75 mL/min/1.73 m^2^ [18]. In our study, all of the participants had been diagnosed with sepsis-induced multiorgan dysfunction. They first met the criteria of circulatory shock refractory to fluid resuscitation and were subsequently treated with CVVH due to one of the following reasons: (1) Azotemia [blood urea nitrogen (BUN) of >80 mg/dL and sCr of >2 mg/dL] with uremic symptoms; (2) Oliguria (urine output of <100 mL for 8 h); (3) Diuretic-refractory fluid overload accompanied by a central venous pressure (CVP) of >12 mmHg or body weight increase >10% from baseline; (4) Treatment-refractory hyperkalemia (serum potassium of >5.5 mmol/L); or (5) Treatment-refractory acidosis (HCO_3_ < 15 mmol/L or pH < 7.2 on arterial blood gas analysis); however, the actual prescription of this treatment was based on the physicians’ clinical judgment. We excluded patients who did not meet the criteria of sepsis-induced MODS, who were treated with RRT modalities other than CVVH, or hemofilters other than AN69 membranes (Gambro Prismaflex M60/M100/M150 set, Baxter, Italy). We also excluded those that were receiving RRT before admission, staying in hospital for less than 48 h, or lacking laboratory data within 24 h of admission and CVVH initiation. The eligible participants were followed up until their death or discharge from hospitalization. According to the recommended level of CRRT intensity by KDIGO guidelines, we categorized the patients that were treated with UFR at 20–25 mL/kg/h for CVVH as the standard UFR group and those that were treated with UFR > 25 mL/kg/h as the high UFR group. Regarding the anticoagulation protocol for CVVH, we injected a bolus of 500 IU of unfractionated heparin pre-filter into a dialysis circuit arterial port, with a continuous infusion of heparin that ranged between 250 and 500 IU/hour. An anticoagulation-free protocol was adopted for patients with active bleeding, severe thrombocytopenia (platelet counts of less than 20,000/uL), coagulopathies, hepatic failure, and those who underwent major surgeries within 3 days.

### 2.3. Data Collection and Follow-Up

We obtained the baseline information of the participants on the index day of hospitalization, including data on demographics (age, sex, baseline renal function, body weight, and body height), medical or surgical hospitalizations, the primary source of infection in sepsis, and comorbidities. The causes of sepsis included pulmonary infections, urinary tract infections, intra-abdominal infections, soft tissue infections, central nervous system infections, and bloodstream infections. Comorbidities included diabetes mellitus (DM), hypertension, liver cirrhosis, coronary artery disease (CAD), congestive heart failure (CHF), chronic obstructive pulmonary disease (COPD), cerebrovascular disease (CVD), chronic kidney disease (CKD), and malignancy. Moreover, we recorded the details of the medical interventions that were performed during the participants’ ICU stay, such as the requirement of diuretics and vasoactive drug support (vasopressors and inotropes), the use of mechanical ventilation (MV), intra-aortic balloon pumps (IABP), and extracorporeal membrane oxygenation (ECMO). The recorded parameters were the UFR and the blood flow setting of CVVH, input and output of fluids in terms of cumulative fluid balance, mean arterial pressure (MAP), the ratio of arterial partial pressure of oxygen to the fraction of inspired oxygen (PaO_2_/FiO_2_), CVP, and the laboratory data within 24 h of CVVH initiation. By the first day of ICU admission and the day of CVVH initiation, we calculated the Acute Physiology and Chronic Health Evaluation II (APACHE II) score to estimate the disease severity. Moreover, we recorded the sequential organ failure assessment (SOFA) score to assess the number and degree of organ dysfunctions. The primary outcome was in-hospital mortality that was defined as any death that occurred during hospitalization. We calculated the survival period from CVVH initiation to mortality (in non-survivors) or hospital discharge (in survivors). If a patient developed a second episode of MODS after ICU discharge, only the first episode was considered.

### 2.4. Statistical Analysis

Descriptive statistics were used to characterize the study cohort. The baseline characteristics were compared between the standard and high UFR groups. For the categorical variables, the χ2 test was performed and for the continuous variables (whether parametric or nonparametric), the independent *t*-test and Mann–Whitney U-test were conducted. Binary logistic regression analysis was employed to assess the association of the pre-specified subgroups and all of the patients with the UFR of CVVH therapy and in-hospital mortality. Next, we used a series of Cox regression models to determine the independent predictors of mortality. The independent variables with *p* ≤ 0.1 on univariate analysis were selected for multivariate analysis. According to the independent risk factors, the subgroups were assessed with the survival curves to evaluate the mortality outcome between the standard and high UFR groups under the modified Kaplan–Meier method. They were also tested with the log-rank statistic. The analyses were performed using SAS software, Version 9.3 of the SAS System (SAS Institute Inc., Cary, NC, USA). Statistical significance was defined as a *p* value of < 0.05.

## 3. Results

### 3.1. Patient Characteristics

The patient enrollment process in our study is depicted in Figure 1.

Of the 334 potentially eligible patients, 266 were enrolled. The standard and high UFR group contained 124 (46.7%) and 142 (53.3%) patients, respectively (Table 1). The mean age of the participants was 66.8 years-old, and the patients were predominantly male (66.9%). The most common comorbidity among all of the patients was hypertension (55.5%), followed by DM (45.7%), CHF (27.6%), CAD (24.5%), advanced CKD (23.0%), and malignancy (22.6%). The mean baseline sCr level was 1.55 mg/dL, with an eGFR of 61.10 mL/min/1.73 m^2^. The baseline characteristics were comparable between the groups, except for the lower percentage of hypertension and body mass index (BMI) in the high UFR group. The proportions of the patients with DM, COPD, CAD, CHF, CVD, and pre-existing CKD were comparable between the groups. Of the 266 patients, 192 (72.2%) were hospitalized in medical ICUs, and the other 74 (27.8%) were in surgical services. Between the standard and higher UFR groups, the distribution of patients who received medical or surgical ICU services was similar. Moreover, the proportions of AKI etiologies were comparable between the groups; the three most common causes of AKI were shock (87.5%), sepsis (72.0%), and nephrotoxicity (11.3%). The mean SOFA and APACHE II scores on the first day of ICU admission were 9.2 and 20.5, respectively. A total of 196 (73.7%) patients died during the index hospitalization, and the mortality rates did not differ significantly between the standard and high UFR groups (75.0% and 72.5%, respectively, *p* = 0.720). A total of 102 (38.3%) patients had a multi-infection focus exceeding two sites. The most common source of infection was the respiratory tract (66.5%), followed by bloodstream (36.1%), urinary tract (34.6%), intra-abdominal organs (10.9%), and cardiovascular system (7.9%). Our analysis showed that the site of the infection does not have a significant impact on the in-hospital mortality in both medical or surgical patients. The lengths of hospitalization and ICU stay were also similar between the two groups.

### 3.2. Clinical Variables and Laboratory Values at the Initiation of CVVH

Table 2 presents the clinical variables and laboratory data of the participants. In the standard and high UFR groups, the mean UFR was 23 and 35 mL/kg/h, respectively. The blood flow of CVVH was set in the range of 150–200 mL/min in both groups (average: approximately 160 mL/min). We observed a significantly shorter mean time interval between ICU admission and the start of CVVH in the high UFR group, with an average of 2.5 days compared with 4.0 days in the standard group (*p* = 0.024). Most of the patients required vasoactive drug support (94%) and MV (89%). However, the mean values of systolic and diastolic blood pressure, MAP, CVP, and percentages of vasoactive drugs use were all similar between the two arms. There were no significant between-group differences in the cumulative fluid balance, urine output, PaO_2_/FiO_2_ ratio, MV, diuretics administration, IABP, and ECMO use at CVVH initiation. Also comparable between the groups were the laboratory data that were collected within 24 h of CVVH initiation, such as those on complete blood count, blood chemistry (e.g., BUN, sCr, electrolytes, lactate, bicarbonate, and pH levels). In addition, the average total SOFA and APACHE II scores at CVVH initiation (10.5 and 23.6, respectively) were comparable between the groups.

### 3.3. Risk of In-Hospital Mortality in the Standard and High UFR Groups

Figure 2 illustrates the risk of in-hospital mortality in the standard and high UFR subgroups. The subgroups were pre-specified according to the presence of shock requiring vasopressor support versus not requiring vasopressor support, the presence of a multi-infection focus exceeding two sites, stratification of the baseline eGFR (≥60 mL/min/1.73 m^2^, 30–60 mL/min/1.73 m^2^, and ≤30 mL/min/1.73 m^2^), presence of oliguria, and APACHE II and SOFA scores at CVVH initiation. Higher mortality rates were noted in individuals who had shock that was requiring vasopressor support, a multi-infection focus, a baseline eGFR of ≤30 mL/min/1.73 m^2^, oliguria, and high APACHE II and SOFA scores at CVVH initiation. Whether the pre-specified subgroups or among all of the patients, no significant differences in the mortality risk that was based on the standard versus high UFR were observed.

### 3.4. Clinical Determinants of Mortality Risk among Patients with Sepsis-Induced MODS

We employed a Cox proportional hazard model to assess the impacts of factors that were associated with mortality in patients with sepsis-induced MODS (Table 3). The characteristics and laboratory data that were collected within 24 h of ICU admission were analyzed, including the age (<65 vs. ≥65 years), BMI (<25 vs. ≥25 kg/m^2^), presence of oliguria, baseline eGFR (≥60 vs. <60 mL/min/1.73 m^2^), hemoglobin level (≥10 vs. <10 g/dL), lactate levels (<4 vs. ≥4 mmol/L), albumin levels (≥3.5 vs. <3.5 g/dL), SOFA scores (<10 vs. 10–14 vs. ≥15), and APACHE II scores (<10, 10–19, 20–29, and ≥30). The factors with *p* < 0.1 in univariate analysis were adjusted in the multivariable regression models. Multivariable analysis revealed that lower baseline eGFRs [<60 mL/min/1.73 m^2^; hazard ratio (HR): 1.77, 95% confidence interval (CI): 1.49–1.92, *p* = 0.017], lower hemoglobin levels (<10 g/dL; HR: 1.53, 95% CI: 1.12–2.03, *p* = 0.012), and higher SOFA scores (≥15; HR: 1.92, 95% CI: 1.31–2.83, *p* < 0.001) were independently associated with a higher mortality risk.

As presented in Figure 3, in the analysis of the subgroup with SOFA scores of ≥15, high UFR was associated with a significantly lower 90-day mortality rate than was the standard UFR (HR: 0.54, 95% CI: 0.36–0.79, *p* = 0.005). However, the survival curves in the subgroup analysis of the baseline eGFR and hemoglobin levels in the two groups were similar.

## 4. Discussion

In this multi-center observational study, among patients with sepsis-induced MODS that were treated with CVVH, variables such as pre-existing CKD, lower hemoglobin levels (<10 g/dL), and higher SOFA scores (≥15) at CVVH initiation were independently associated with in-hospital mortality. The binary logistic regression analysis revealed no significant CVVH-based differences in the in-hospital mortality rates between the standard and high UFR groups. However, we observed that in the subgroup of patients with SOFA scores of ≥15, during CVVH that was applied with AN69 membranes, a high UFR (≥25 mL/kg/h) reduced the risk of 90-day mortality compared with the standard UFR (20–25 mL/kg/h). Although the prognosis of patients with sepsis-induced MODS is generally poor, our study presents an attemptable treatment to reduce mortality in individuals with severe illness and organ failure.

Sepsis can induce an imbalance between the oxygen supply and demand, leading to tissue hypoxia and a series of cellular derangements (e.g., lactic acid production and microcirculatory and mitochondrial dysfunction), all of which may lead to shock and MODS [19]. Moreover, endotoxins and an exacerbated inflammatory immune response cause tissue damage and the acute release of pro-inflammatory cytokines, such as TNF-α, IL-1, and IL-6, resulting in hypercatabolism, proteolysis, lipolysis, and insulin resistance [20]. In the early 1990s, Bellomo et al. indicated that CRRT therapy attenuated the progression of AKI and removed inflammatory cytokines from the circulation of patients with sepsis [6]. Servillo et al. reported that the immunomodulatory effect of CVVH could be attained by using a high UFR of 60 mL/kg/h in critically ill patients with AKI that were accompanied by severe sepsis or septic shock [21]. Although modern high-flux membranes with an average cutoff of approximately 30–40 kD should be capable of eliminating inflammatory cytokines by convection, some have questioned whether the amount that is removed is clinically significant when considering the high production speeds and turnover rates of the mediators [22]. Moreover, De Vriese et al. demonstrated that CVVH removes cytokines from the circulation in patients with sepsis through adsorption within the first hour after the placement of a new hemodialyzer into the circuit [23]. However, the filter membranes rapidly became saturated with cytokines, and the anti-inflammatory mediators were removed to the same extent as the inflammatory cytokines, which may explain the lack of survival benefits of high-intensity CRRT in previous studies [14,15]. Our findings suggest that septic patients with high SOFA scores who are treated with AN69 membranes at higher UFRs in CVVH have improved 90-day survival rates. Our data parallel those of prospective studies that suggest that high ultrafiltration volumes of CVVH may improve ICU survival in the treatment of patients with severe sepsis [13,24]. In addition, Ronco et al. proposed the peak concentration hypothesis, which holds that CVVH, particularly at high volumes, can remove the unbound inflammatory mediators and cytokines from blood compartments, thereby restoring immune homeostasis between the infected tissues and the circulation [25]. Although we did not routinely measure the cytokine changes during CVVH, the cytokine adsorptive characteristics of the AN69 membrane in the treatment of sepsis have been recognized in previous studies [23,24,25]. Therefore, prescribing high UFR and cytokines adsorption filters during CVVH for sepsis-induced MODS is reasonable under the premise that a hypermetabolic state contributes to higher protein waste production and caloric needs in critically ill patients.

Our data revealed that pre-existing CKD, hemoglobin levels of <10 g/dL, and high SOFA scores strongly predicted mortality in the ICU setting. Renal dysfunction increases the risk of mortality among critically ill patients, and the development of AKI in patients with CKD exerts negative effects on the long-term mortality and dialysis dependence after hospital discharge [26,27,28]. The effect of high-UFR CVVH on the long-term mortality and renal recovery among sepsis survivors requires further investigation. The current Surviving Sepsis guidelines recommends aiming for a hemoglobin level of 7–9 g/dL in ICU patients without myocardial ischemia, severe hypoxemia, acute hemorrhage, or ischemic CAD [29]. By contrast, our multivariate Cox regression analysis revealed that a hemoglobin level of <10 g/dL was independently associated with mortality. This discrepancy may be attributed to the participants’ high oxygen and transfusion demands, which is, in turn, due to the severity of their conditions; almost all of the patients with MODS develop anemia during their ICU stay. The complex pathophysiological mechanisms of MODS, which involve microcirculation alterations, reticuloendothelial system activation, and cytokine storms, lead to red blood cell destruction and reduced bone marrow erythropoiesis [30]. Furthermore, blood loss due to trauma, surgical procedures, clotting on extracorporeal circuits, or gastrointestinal bleeding are common causes of anemia in critically ill patients. Therefore, we suggest maintaining higher-than-recommended hemoglobin levels in ICU patients with severe sepsis.

Developed by Vincent et al., the SOFA score is effective in predicting the ICU mortality based on the assessment of the severity of dysfunction in organs (e.g., the lungs, blood, liver, heart, and kidneys) and in the central nervous system [31]. Ferreira et al. sequentially measured the SOFA scores in 352 critically ill patients during the first 96 h of admission, concluding that patients whose SOFA scores increased in the first 48–96 h were associated with a significantly higher mortality rate than the patients whose SOFA scores decreased during the same interval [32]. In our patients, the mean SOFA scores were 9.2 on the first day of ICU admission and 10.5 at CVVH initiation, separated by a time interval of approximately three days. This may explain the higher mortality rate of 73% in our cohort compared with the ICU mortality rate of 40% that was reported in a systematic review and meta-analysis [33]. In our subgroup of patients with SOFA scores of ≥15, the 83% all-cause mortality rate aligns with the survival rates that were estimated from the SOFA scores. No significant difference in mortality between the standard and high UFR groups was observed. However, the subgroup analysis addressed the importance of higher UFRs in CVVH. Specifically, increasing the 90-day survival rates in critically ill patients with high SOFA scores could be achieved under high UFR settings.

This study has some limitations. First, all of the participants were Taiwanese; thus, our results may not be applicable to other ethnic groups. Second, we must acknowledge that the factors interrupting the CVVH, such as extracorporeal circuit clotting, a decline in filter efficacy, catheter problems, and external ICU procedures, would reduce the treatment time. However, we could not provide a detailed treatment course for each patient in this registry-based study. Moreover, due to the lack of clinical data registries including pre-dilution and post-dilution, frequency of filter changes, and duration of CVVH, we could not assess the delivered dosage of CVVH for each patient. However, the CVVH protocol in each participating site strictly followed the clinical practice guidelines, and we used the guideline recommended UFR setting of 25 mL/kg/h as the cutoff. Third, this was a registry-based retrospective design, and is, therefore, missing data while also being subject to potential selection bias. Although our data were adjusted through a multivariate analysis, the potential for bias due to an unmeasured confounder remains. Finally, the observational nature of our study precludes the determination of causality. Nevertheless, this is the first multicenter study in Taiwan to investigate the effect of CRRT on the survival benefits in ICU patients with sepsis-induced MODS. The recruitment of critically ill patients into clinical trials is challenging. Our study constitutes a safe and applicable attempt at reducing mortality in critically ill patients with sepsis-induced MODS.

## 5. Conclusions

Among patients with sepsis-induced MODS, pre-existing comorbid CKD, hemoglobin levels of <10 g/dL, and SOFA scores of ≥15 were independently associated with mortality. In patients with SOFA scores of ≥15, a high UFR setting with AN69 filter membranes during CVVH were associated with a higher 90-day survival rate than was a standard UFR setting. Additional prospective clinical studies are required to validate our findings.

## Figures and Tables

**Figure 1 membranes-11-00837-f001:**
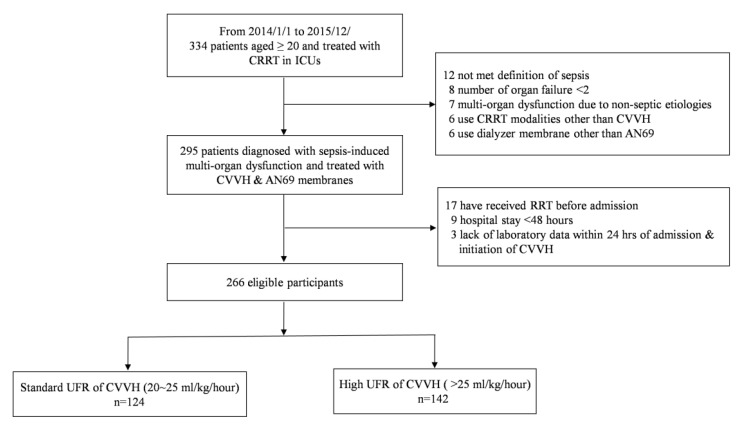
Study flowchart. This multi-center, registry-based cohort consisted of 334 adults that were treated with CRRT between 2014 and 2015. The final sample comprised of 266 patients with sepsis-induced MODS who were treated with CVVH. According to their prescribed CVVH settings, they were assigned to either the standard UFR group (20–25 mL/kg/h; *n* = 124) or the high UFR group (>25 mL/kg/h; *n* = 142). Abbreviations: CRRT, continuous renal replacement therapy; ICU, intensive care units; CVVH, continuous venovenous hemofiltration; RRT, renal replacement therapy; UFR, ultrafiltration rate.

**Figure 2 membranes-11-00837-f002:**
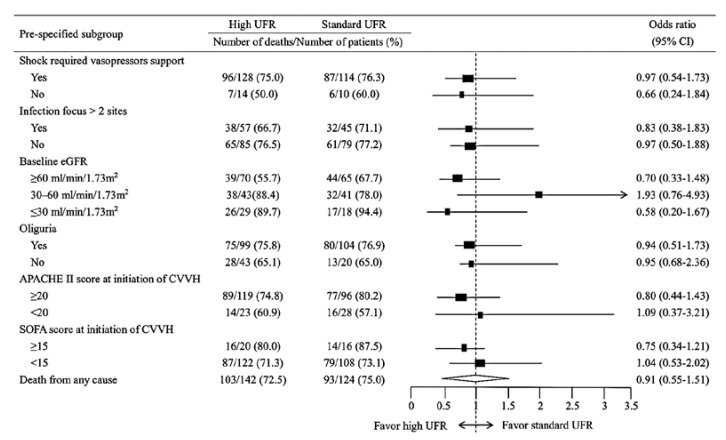
Forest plot of differences in the odds ratio estimates for mortality that are based on standard and high UFRs in the overall cohort and the pre-specified subgroups. Overall and in both subgroups, the risks for in-hospital mortality among patients with sepsis-induced MODS with standard versus high UFR were comparable. The black boxes represent the odds ratios, and the horizontal lines indicate the 95% confidence intervals. The vertical dashed line represents an odds ratio of 1.0. The diamond represents the pooled odds ratio, and its lateral tips represent the confidence intervals. Abbreviations: UFR, ultrafiltration rate; eGFR, estimated glomerular filtration rate; CVVH, continuous venovenous hemofiltration; APACHE, Acute Physiology and Chronic Health Evaluation; SOFA, sequential organ failure assessment.

**Figure 3 membranes-11-00837-f003:**
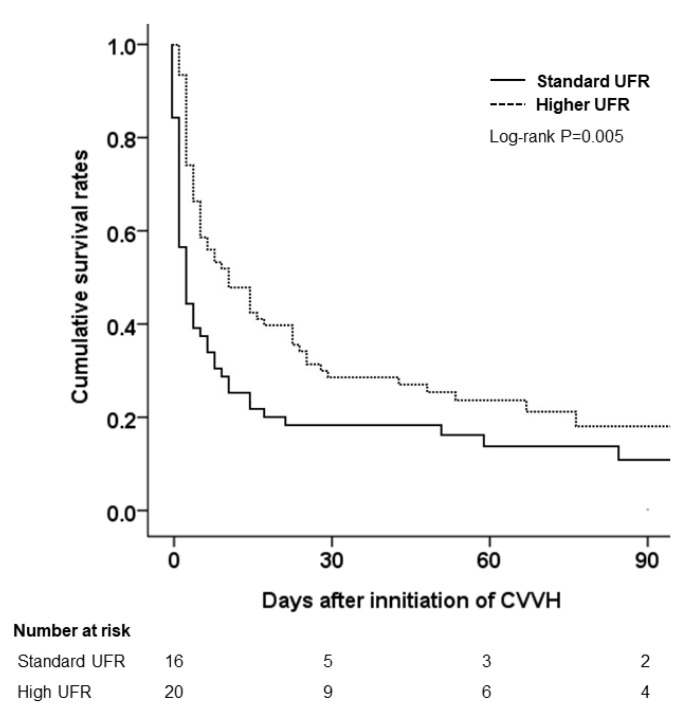
Kaplan–Meier curves of the 90-day mortality that was based on standard and high UFRs in a subgroup of patients with SOFA scores of ≥15. Abbreviations: UFR, ultrafiltration rate; CVVH, continuous venovenous hemofiltration.

**Table 1 membranes-11-00837-t001:** Demographic data of patients that were receiving continuous venovenous hemofiltration at standard versus high ultrafiltration rates.

	Standard UFR (*n* = 124)	High UFR (*n* = 142)	*p* Value
Background			
Male, *n* (%)	83 (66.9)	95 (66.9)	0.383
Age (years)	68.09 ± 14.12	65.97 ± 15.52	0.264
BMI (kg/m^2^)	25.83 ± 4.50	23.66 ± 4.58	<0.001 *
Baseline sCr (mg/dL)	1.55 ± 1.36	1.55 ± 1.36	0.992
Baseline eGFR (mL/min/1.73 m^2^) ^†^	65.32 ± 32.89	63.02 ± 38.41	0.813
Comorbidity			
Hypertension, *n* (%)	80 (64.5)	70 (49.3)	0.015 *
Diabetes mellitus, *n* (%)	62 (50.0)	12461 (43.0)	0.286
Liver cirrhosis, *n* (%)	16 (12.9)	21 (14.8)	0.665
Coronary artery disease, *n* (%)	30 (24.2)	35 (24.6)	0.982
Congestive heart failure, *n* (%)	34 (27.4)	39 (27.5)	0.998
COPD/Chronic lung disease, *n* (%)	7 (5.6)	12 (8.5)	0.425
Cerebrovascular disease, *n* (%)	15 (12.1)	16 (11.3)	0.965
Advanced CKD ^‡^, *n* (%)	32 (25.8)	30 (21.1)	0.434
Malignancy, *n* (%)	24 (19.4)	36 (25.4)	0.309
Charlson comorbidity index	6.42 ± 3.25	62.23 ± 3.00	0.600
Primary ICU service received			
Medical, *n* (%)	94 (75.8)	98 (69.0)	0.248
Surgical, *n* (%)	30 (24.2)	44 (31.0)	
Etiology of acute kidney injury			
Shock, *n* (%)	112 (90.3)	122 (85.9)	0.310
Sepsis, *n* (%)	91 (73.4)	101 (71.1)	0.699
Nephrotoxins, *n* (%)	18 (14.5)	13 (9.2)	0.163
Hepatorenal, *n* (%)	7 (5.6)	12 (8.5)	0.425
Cardiorenal, *n* (%)	4 (3.2)	5 (3.5)	0.753
Rhabdomyolysis, *n* (%)	5 (4.0)	5 (3.5)	0.702
Others, *n* (%)	4 (3.2)	5 (3.5)	0.753
Outcomes			
Length of hospital stay (days)	31.23 ± 32.37	30.34 ± 32.11	0.808
Length of ICU stay (days)	16.71 ± 19.71	16.36 ± 19.84	0.877
Death or critical AAD, *n* (%)	93 (75.0)	103 (72.5)	0.720

^*^ *p* < 0.05. ^†^ estimated using the Modification of Diet in Renal Disease equation. ^‡^ defined as baseline estimated glomerular filtration rates less than 30 mL/min/1.73m^2^. Abbreviations: UFR, ultrafiltration rate; BMI, body mass index; sCr, serum creatinine; eGFR, estimated glomerular filtration rate; COPD, chronic obstructive pulmonary disease; CKD, chronic kidney disease; ICU, intensive care unit; AAD, against-advise discharge.

**Table 2 membranes-11-00837-t002:** Characteristics at the initiation of continuous venovenous hemofiltration in the standard and high ultrafiltration rate groups.

	Standard UFR (*n* = 124)	High UFR (*n* = 142)	*p* Value
Clinical variables			
UFR of CVVH (mL/kg/h)	22.99 ± 1.97	35.14 ± 8.69	<0.001 *
CVVH blood flow (mL/min)	159.8 ± 23.42	158.6 ± 24.67	0.735
Interval between admission and CVVH initiation (days)	4.13 ±1.80	2.57 ± 0.38	0.024 *
Cumulative fluid balance (kg)	4.41 ± 5.85	4.23 ± 8.08	0.870
Urine output (mL/kg/hr)	0.24 ± 0.32	0.30 ± 0.53	0.242
Body temperature (°C)	36.65 ± 1.35	36.57 ± 1.44	0.657
Systolic blood pressure (mmHg)	82.53 ± 11.99	84.34 ± 13.17	0.838
Diastolic blood pressure (mmHg)	42.38 ± 10.21	39.06 ± 8.79	0.571
MAP (mmHg)	55.76 ± 10.83	54.15 ± 12.72	0.747
CVP (mmHg)	15.41 ± 6.24	16.50 ± 6.52	0.244
Vasoactive drug use, *n* (%)	115 (92.7)	135 (95.1)	0.720
PaO_2_/FIO_2_ ratio (mmHg)	218.21 ± 139.03	240.52 ± 144.21	0.222
MV use, *n* (%)	113 (91.1)	125 (88.0)	0.389
Diuretic use, *n* (%)	72 (58.1)	83 (58.5)	0.932
IABP use, *n* (%)	13 (10.5)	11 (7.7)	0.429
ECMO use, *n* (%)	12 (9.7)	15 (10.6)	0.752
Indications for CVVH			
Azotemia (BUN > 80 and sCr of >2 mg/dL) with uremic symptoms	59 (47.58)	47 (33.10)	0.252
Oliguria (UO < 100 mL for 8 h)	111 (89.52)	121 (85.21)	0.716
Diuretic-refractory fluid overload (CVP > 12 mmHg or BW increase >10%)	106 (85.48)	117 (82.39)	0.406
Treatment-refractory hyperkalemia (serum potassium >5.5 mmol/L)	63 (50.81)	57 (40.14)	0.195
Treatment-refractory acidosis (HCO_3_ < 15 mmol/L or pH < 7.25)	95 (76.61)	103 (72.54)	0.628
Laboratory data			
Lactate (mmol/L)	7.63 ± 5.71	7.26 ± 6.53	0.682
Albumin (g/dL)	2.80 ± 0.67	2.73 ± 0.69	0.464
White blood cell (×10^3^/μL)	15.57 ± 10.63	14.21 ± 12.32	0.357
Hemoglobin (g/dL)	10.15 ± 2.71	10.11 ± 2.38	0.897
Platelet (×10^3^/μL)	136.83 ± 100.58	133.67 ± 99.53	0.804
Arterial blood pH	7.32 ± 0.12	7.33 ± 0.12	0.329
Bicarbonate (mmol/L)	16.65 ± 5.11	17.15 ± 5.56	0.460
Sodium (mmol/L)	140.9 ± 8.00	139.6 ± 10.09	0.280
Potassium (mmol/L)	4.61 ± 1.15	4.51 ± 1.09	0.453
Blood urea nitrogen (mg/dL)	63.65 ± 38.99	66.82 ± 43.29	0.546
sCr (mg/dL)	3.47 ± 1.88	3.60 ± 2.18	0.612
Severity of illness			
SOFA score	10.37 ± 5.89	10.97 ± 6.28	0.525
APACHE II score	22.24 ± 8.13	23.39 ± 7.85	0.203

* *p* < 0.05. Abbreviations: UFR, ultrafiltration rate; CVVH, continuous venovenous hemofiltration; MAP, mean arterial pressure; PaO_2_/FiO_2_, the ratio of arterial oxygen partial pressure to fractional inspired oxygen; CVP, central venous pressure; MV, mechanical ventilation; IABP, intra-aortic balloon pump; ECMO, extracorporeal membrane oxygenation; BUN, blood urea nitrogen; sCr, serum creatinine; BW, UO, urine output; BW, body weight; HCO_3_, bicarbonate; SOFA, sequential organ failure assessment; APACHE, Acute Physiology and Chronic Health Evaluation.

**Table 3 membranes-11-00837-t003:** Univariate and multivariate Cox regression analysis of variables associated with in-hospital mortality.

Variables	Univariate Analysis		Multivariate Analysis ^†^	
	HR (95% CI)	*p* Value	HR (95% CI)	*p* Value
Age, years				
<65	1			
≥65	0.9 (0.66–1.22)	0.498		
BMI, kg/m^2^				
<25	1			
≥25	1.31 (0.96–1.79)	0.584		
Oliguria				
No	1			
Yes	1.07 (0.75–1.54)	0.708		
Baseline eGFR, mL/min/1.73 m^2^				
≥60	1			
<60	1.93 (1.48–2.88)	0.005 *	1.77(1.49–1.92)	0.017 *
Hemoglobin, g/dL				
≥10	1			
<10	1.43 (1.04–1.96)	0.029 *	1.53 (1.10–2.13)	0.012 *
Lactate, mmol/L				
<4	1			
≥4	1.56 (1.07–2.27)	0.021 *	1.28 (0.86–1.89)	0.227
Albumin, g/dL				
≥3.5	1			
<3.5	1.74 (0.98–3.06)	0.057	1.43 (0.80–2.55)	0.225
SOFA score				
<10	1			
10–14	1.83 (0.92–3.66)	0.087		
≥15	2.85 (1.44–5.65)	0.003 *	1.92 (1.31–2.83)	<0.001 *
APACHE II score				
<10	1			
10–19	0.92 (0.56–1.50)	0.737		
20–29	0.80 (0.59–1.08)	0.142		
≥30	1.32 (0.97–1.81)	0.076 *	1.16 (0.84–1.60)	0.360

* *p* < 0.05. ^†^ All factors with a *p* value of < 0.1 in univariate analysis were included in the Cox multivariate analysis. Abbreviations: HR hazard ratio; CI, confidence interval; BMI, body mass index; UFR, ultrafiltration rate; CVVH, continuous venovenous hemofiltration; SOFA, sequential organ failure assessment; APACHE, Acute Physiology and Chronic Health Evaluation.

## Data Availability

The datasets generated for this study are available upon request to the corresponding author.

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
