# Peer review of "AN69 Filter Membranes with High Ultrafiltration Rates during Continuous Venovenous Hemofiltration Reduce Mortality in Patients with Sepsis-Induced Multiorgan Dysfunction Syndrome"

_membranes, 2021, doi:10.3390/membranes11110837_

Round 1

Reviewer 1 Report

In this retrospective work, the authors evaluated the use of polyacrylonitrile filter membrane (AN69) with different ultrafiltration rates (UFR) during continuous veno-venous hemofiltration for treatment of sepsis-induced multiorgan dysfunction syndrome. They investigated the effects of two different rates of UFR on mortality: 20-25 mL/kg/h as standard UFR group and >25 mL/kg/h as high UFR group. The number of subjects examined is quite high (over 250 patients from 10 medical centers). The Authors conclude that high UFR setting may reduce mortality risk in septic patients.

The work is well written. I have the following comments:

1) Between the two study groups, in which percentage was there a medical or surgical problem as the cause of hospitalization? And, in case of surgical alterations were there any differences in mortality based on the origin of the sepsis? (e.g. lung to abdomen?)

2) It would be advisable to indicate in Table 2 the values of pH, C-reactive protein and temperature at the beginning of the treatments

3) Were there any interruptions observed in the two different treatment modalities? What was the heparin dosage? How was the anticoagulation regimen performed considering that these patients can undergo bleeding episodes?

Reviewer 2 Report

General comments

In this manuscript, the authors described the effect of high ultrafiltration rate in CVVHF using AN69 filter from multicenter retrospective analysis. The manuscript contains some novel factor, however, we should address some technical matter.

Comments

  1. This study is a retrospective cohort that registered clinical reality and is a non-RCT. Since the decision of whether to use high or standard UFR is dependent on the attending physician, there is a possibility that the grouping may be arbitrary. Could it be that the patients who chose high UFR were able to do so because of their stable circulatory status? Isn't this the reason for the difference in primary endpoints?
  2. A detailed description of the indication is needed. Were all patients treated with AN69 membranes as an indication for AKI? Or did they include non-renal indication cases of sepsis alone?
  3. Was there a difference in the degree of cytokines due to differences in adsorption between groups?
  4. Comorbidity should be assessed with the charlson comorbidity index.
  5. Ronco's high UFR is defined as 35 ml/kg/h or higher, but can the authors defined 25 ml/kg/h as high UFR?

Round 2

Reviewer 2 Report

The authors responded appropriately to the revision of the paper. The content has been improved and is useful information for readers. Thank you for your efforts.